# Just Label What You Need: Fine-Grained Active Selection for P&P through Partially Labeled Scenes

**Sean Segal**[12*]**, Nishanth Kumar**[3*]**, Sergio Casas**[12]**, Wenyuan Zeng**[12]**,
Mengye Ren**[12]**, Jingkang Wang**[12]**, Raquel Urtasun**[12]
[†] Waabi[1], University of Toronto[2], MIT CSAIL[3]
{ssegal, scasas, wzeng, mren, jwang, urtasun}@waabi.ai, njk@csail.mit.edu

**Abstract:** Self-driving vehicles must perceive and predict the future positions of nearby actors to avoid collisions and drive safely. A deep learning module is often responsible for this task, requiring large-scale, high-quality training datasets. Due to high labeling costs, active learning approaches are an appealing solution to maximizing model performance for a given labeling budget. However, despite its appeal, there has been little scientific analysis of active learning approaches for the perception and prediction (P&P) problem. In this work, we study active learning techniques for P&P and find that the traditional active learning formulation is ill-suited. We thus introduce generalizations that ensure that our approach is both cost-aware and allows for fine-grained selection of examples through partially labeled scenes. Extensive experiments on a real-world dataset suggest significant improvements across perception, prediction, and downstream planning tasks.

## 1 Introduction

For self-driving vehicles to safely plan a route, they must perceive nearby actors and predict their future locations. In a self-driving stack, a learned perception and prediction (P&P) model is responsible for this task, taking raw sensor data as input and producing object detections and future predictions. These models require large-scale, high-quality training datasets due to the high dimensional sensor inputs and long tail of future outcomes. While self-driving companies collect massive amounts of data from real-world driving, annotating the data remains a major bottleneck. Furthermore, some of the data may be less interesting for model training – e.g., a prediction dataset with many parked vehicles is less informative than one with highly interactive, moving actors. Therefore, the choice of which examples to label is crucial to maximize performance for a given budget.

Given a particular model, we seek to determine the examples most likely to improve performance when labeled. This problem is well-studied in the field of active learning and recent work has shown impressive performance gains over random selection in many tasks, including image classification, semantic segmentation and 2D object detection [1, 2, 3]. Active learning presents a promising framework to employ in real-world self-driving development, where models can continually improve as new batches of examples are selected iteratively for labeling (see Figure 1).

Despite the appeal of active learning, few approaches have been developed for self-driving. Scientific analysis is limited to object detection [4, 5], with no approaches designed for P&P. When applying active selection to this task, we find the traditional formulation to be ill-suited. First, while approaches typically assume fixed labeling costs per example, annotation costs can vary drastically as they depend on the number of actors present. Furthermore, the spatial label structure can be exploited to support partial labeling, enabling fine-grained active selection. Specifically, with small modifications, P&P models can be trained from partial supervision. This allows the active learner to select specific actors in a scene without requiring the remaining actors to be selected, as they may be

---

[*]Equal Contribution
[†]This work was done by all authors while at Uber ATG

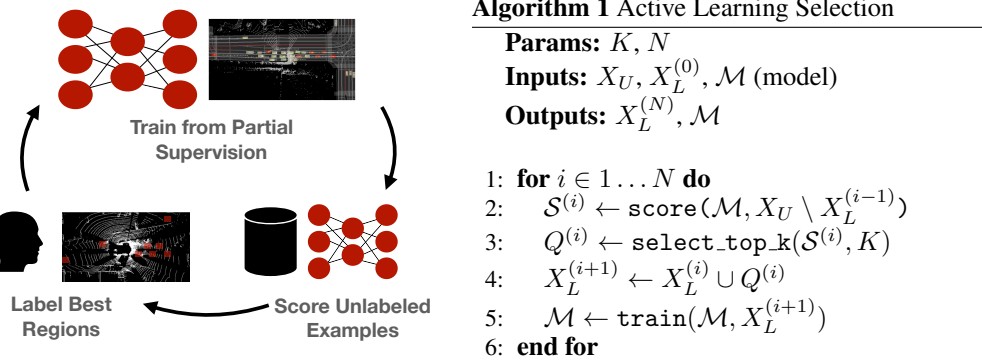

Figure 1: (**Left**) Overview of fine-grained active selection. (**Right**) Selection Algorithm.



**Algorithm 1** Active Learning Selection

**Params:** $K$, $N$
**Inputs:** $X_U$, $X_L^{(0)}$, $\mathcal{M}$ (model)
**Outputs:** $X_L^{(N)}$, $\mathcal{M}$

1: **for** $i \in 1 \ldots N$ **do**
2: $\quad \mathcal{S}^{(i)} \leftarrow \texttt{score}(\mathcal{M}, X_U \setminus X_L^{(i-1)})$
3: $\quad Q^{(i)} \leftarrow \texttt{select\_top\_k}(\mathcal{S}^{(i)}, K)$
4: $\quad X_L^{(i+1)} \leftarrow X_L^{(i)} \cup Q^{(i)}$
5: $\quad \mathcal{M} \leftarrow \texttt{train}(\mathcal{M}, X_L^{(i+1)})$
6: **end for**



uninteresting for model improvement. With these differences in mind, we introduce a fine-grained, cost-aware selection along with specific scoring criteria for active selection in the P&P setting.

We leverage a real-world, large-scale dataset to analyze the effects of partial labeling and active selection for P&P. First, we study models trained on partial supervision, without active selection, and observe improved performance from sparser labeling. Next, we analyze the added benefits of fine-grained active selection and observe significant gains compared to traditional approaches. Further analysis shows the gains are most signifiant on rare events, translating to improvements in downstream planning performance. All together, our analysis suggests that the dominant paradigm of labeling entire self-driving scenes is not the most efficient use of a fixed labeling budget and that more fine-grained active selection may be required to most effectively select examples for labeling.

## 2 Related Work

**Perception and Prediction:** While perception and prediction have traditionally been handled separately, [6, 7] introduce models to jointly perform both tasks, improving performance and efficiency. Among the exciting progress made in both tasks over recent years, most relevant to our work are improvements in prediction representations, allowing models to better characterize uncertainty. Examples of representations include trajectories [7, 8], probabilistic occupancy maps [9, 10], Gaussian mixtures [11], implicit latent variable models [12], and auto-regressive models [13].

**Active Learning:** We focus on pool-based active learning, in which new training examples are queried from a large, unlabeled pool [14]. One class of approaches seeks to characterize model uncertainty, measured via model disagreement [15, 16], entropy [17, 18], a learned loss prediction [1], or a discriminator score [19], and select examples with high uncertainty for labeling. While often effective, uncertainty based approaches are prone to selecting a subset of similar examples when computational constraints require large batches of examples to be selected before retraining. This motivates diversity-based approaches [20, 21, 22, 23] which seek to find a representative subset of the unlabeled pool. [3, 24, 25] introduce approaches which balance both uncertainty and diversity in the selection process. Most related to our domain are [4, 5, 26, 27, 28] which study active learning approaches for object detection. While most approaches assume fixed labeling costs per example, [29, 30] have explored explicitly modeling individual labeling costs as part of the selection process. Leveraging partially labeled data for fine-grained active selection has been explored in semantic segmentation [31, 32] and more generally, in the context of structured prediction problems [33].

**Dataset Selection:** Many self-driving datasets select examples manually [34], randomly, or via hardcoded rules. Argoverse describes rules-based criteria to mine interesting trajectories for prediction [35]. More recently, [36] proposed a set of complexity measures for dataset selection. Finally, [37] proposed tagging attributes of self-driving scenes, enabling retrieval for dataset curation.

## 3 Active Learning for P&P

Given the high labeling costs of P&P datasets, budgets can be spent more efficiently by intelligently selecting examples for labeling. Active learning offers a promising solution, selecting examples

believed most likely to improve model performance. In this section, we review traditional pool-based active learning, where examples are iteratively selected from an unlabeled pool to build a high-quality labeled dataset. Then, we address shortcomings in the P&P setting by introducing a new paradigm which is both cost-aware and enables fine-grained selection through partially labeled scenes, providing the flexibility to ensure budgets are spent effectively. Finally, we provide concrete selection criteria used within our framework to optimize the model's performance.

## 3.1 Traditional Active Learning

Self-driving companies can collect large-scale unlabeled real-world data when operating their vehicles. Our goal is to select the best subset to label to improve model performance. We assume access to a large, unlabeled pool of examples, $X_U$, and an initial subset of labeled examples, $X_L^{(0)}$. Each example $\mathbf{x} \in X_U$ represents an input to our model $f(\mathbf{x})$ and if selected, a labeling oracle returns the ground truth supervision, $\mathbf{y} = L(\mathbf{x})$. In the P&P setting, inputs $\mathbf{x}$ represent raw sensor observations and HD Maps, and labels $\mathbf{y}$ represent actor bounding boxes at the current timestep and for the prediction horizon of $T$ seconds. In each active learning iteration, we select a subset from the remaining unlabeled examples, $Q^{(i)} \subset X_U \setminus X_L^{(i-1)}$, query the labeling oracle, and add the examples to our labeled set. With each iteration, the model is retrained or fine-tuned with the latest dataset.

Traditionally, the active learner will select a fixed number of examples at each iteration, $|Q^{(i)}| = K$. This implicitly assumes that each example $\mathbf{x} \in X_U$ can be labeled for the same cost, an assumption clearly violated in the P&P setting, which we will relax in the next section. While a variety of approaches have been studied for active selection, we focus on methods which produce a scalar score for each example, $S(\mathbf{x}) \in \mathbb{R}$. Scores represent some notion of informativeness where highly scored examples are believed to be most likely to improve model performance. For example, measures of model uncertainty, such as entropy, are commonly used (see Section 3.3 for concrete scoring functions for P&P). As different models may benefit from different types of examples, most scoring approaches depend on the model's current state. After scores have been computed for the remaining unlabeled examples, the top $K$ examples can be selected for labeling. This process repeats for $N$ active learning iterations and is summarized in Algorithm 1.

## 3.2 Fine-Grained Cost-Aware Active Learning

In this section, we generalize two aspects critical to the P&P setting, allowing for variable labeling costs and fine-grained selection through partial supervision.

**Cost-Aware Active Learning:** As a self-driving vehicle operates, the surrounding environment will change, leading to scenes with different labeling costs. Crowded scenes can contain hundreds of actors, which are each traditionally labeled with a precise bounding box. Sparser scenes, on the other hand, can be labeled with little manual effort. To account for these differences, we explicitly model the cost to label each example, $C(\mathbf{x})$. At each iteration, rather than select a fixed $K$ examples, the learner is instead given a fixed budget $B$, which cannot be exceeded, $\sum_{\mathbf{x} \in Q^{(i)}} C(\mathbf{x}) \leq B$.

This formulation is a generalization of the previous setting, which can be recovered by setting $C(\mathbf{x}) = 1$ for all examples and $B = K$. In practice, labeling cost for P&P examples can be accurately modeled as a linear function of the number of actors in the scene as most annotation time is spent drawing detailed bounding boxes for each actor. This new formulation requires modifications to our selection algorithm, since high scoring examples $S(\mathbf{x})$ may also have high costs $C(\mathbf{x})$. Therefore, rather than sorting by score, we can select examples with the highest value, $V(\mathbf{x}) = \frac{S(\mathbf{x})}{C(\mathbf{x})}$.

Since the cost of labeling a region is unknown before labeling, we approximate it using the number of detections after non-maximum suppression (NMS) [38]. After labeling, the true cost is known, and active selection can continue iteratively until the budget is reached. Since examples in the P&P setting represent large scenes, scores $S(\mathbf{x})$ and costs $C(\mathbf{x})$ can vary significantly as scenes can have few to many actors, each contributing to the total score and cost. As a consequence, coarse-grained scoring will be suboptimal as scenes may contain regions with high score and low cost (e.g., a single

car performing a rare U-Turn) and other regions with low score and high cost (e.g., a parking lot filled with many static vehicles). This motivates the need for more fine-grained scoring and selection. Next, we describe modifications to support partially labeled scenes, which will enable fine-grained selection for better performance in the cost-aware active learning setting.

**Partially Labeled Scenes:** We generalize the labeling process to allow for partial labeling. Along with the added flexibility, this setting is also realistic in practice. Even as entire scenes are labeled today, annotation platforms often decompose work into subtasks, which can be more easily distributed and validated across a labeling team. As a simple extension, platforms could support querying labels for only particular regions. To support partial labels, we redefine an example $\mathbf{x}_R$ as the scene augmented with a labeling region $R$, $\mathbf{x}_R = (\mathbf{x}, R)$. Given the set of labels for the entire scene $\mathbf{y} = L(\mathbf{x})$ and a region $R$, each actor's bounding box label $\mathbf{y}_i$ will either be fully contained in $R$, completely outside of $R$, or partially inside of $R$. For simplicity, we assume that if **any** part of the bounding box $\mathbf{y}_i$ of an actor is inside $R$ then it will be provided as a label. In practice, this translates to labelers annotating all actors, even those that are only partially visible in the labeling region. More formally, the labeling oracle returns labels for an example $\mathbf{x}_R$, $L(\mathbf{x}_R) = \{\mathbf{y}_i : \mathbf{y}_i \in R \text{ and } \mathbf{y}_i \in L(\mathbf{x})\}$.

**Training from Partial Supervision:** We adapt training to support partial supervision by applying the loss only on the labeled region, $R$. Importantly, we do not alter the network input $\mathbf{x}$, since we do not want to bias the network by changing input statistics. Therefore, the forward pass remains unchanged, $\hat{\mathbf{y}} = f(\mathbf{x})$. To compute the loss, we only consider the labels that we have received in $R$,

$$\mathcal{L}(\mathbf{y}, \hat{\mathbf{y}}, R) = \ell_B(R) + \sum_{\mathbf{y}_i \in R} \ell_P(\mathbf{y}_i, \hat{\mathbf{y}}_i) \; . \tag{1}$$

Here, $\ell_P(\cdot, \cdot)$ represents traditional multi-task perception and prediction losses applied over the positive examples in $R$ and $\ell_B(R)$ represents a "background" loss which encourages the network not to output detections for negative regions in $R$. For example, in our experiments, $\ell_P$ includes a probabilistic prediction loss, a bounding box regression loss and cross-entropy on positive examples, whereas $\ell_B(R)$ represents the hard negative mining loss, sampling only negative anchors from $R$.

**Fine-Grained Selection:** Without restrictions on $R$, there are infinite regions to consider for a given scene. Therefore, in order to efficiently score and select regions for labeling, we consider the set obtained by discretizing the entire scene into a rectangular grid. Specifically, we divide each example $\mathbf{x}$ into $HW$ non-overlapping regions, $\mathbf{x}_R = (\mathbf{x}, R_{h,w})$. By setting $H = W = 1$, we obtain a single region for each scene and recover the original formulation. As $H$ and $W$ increase, candidate regions become smaller, providing the learner more fine-grained precision for selection. Most steps of the selection process can remain unchanged. Scoring functions now operate over examples augmented with regions, $S(\mathbf{x}_R)$ returning a score that only considers network predictions in $R$. Similarly, only the cost of labeling the queried region $C(\mathbf{x}_R)$ is incurred when selecting $\mathbf{x}_R$.

As regions sizes shrink, we observe that the active learner is more likely to select a large number of scenes, each labeled with very sparse supervision. While this dataset would contain many interesting actors, we also find this unconstrained selection results in significantly longer training times as examples are less densely labeled. Additionally, we observe training instabilities due to the imbalances between the amount of supervision available for each example. To alleviate these issues, we introduce a sparsity regularizer, which requires that the active learner select a minimum number of positive examples $M$ for any selected scene. Therefore, letting $P(\mathbf{x}_R)$ represent the number of positive examples in an example, our new formulation can be summarized by the following optimization problem solved by the learner at each iteration,

$$\max_{Q^{(i)}} \sum_{\mathbf{x}_R \in Q^{(i)}} S(\mathbf{x}_R) \quad \text{s.t.} \sum_{\mathbf{x}_R \in Q^{(i)}} C(\mathbf{x}_R) \leq B \text{ and } P(\mathbf{x}_R) \geq M \; \forall \, \mathbf{x}_R \in Q^{(i)} \; . \tag{2}$$

We solve this optimization greedily by first selecting the highest scoring scene remaining, then selecting the highest regions in the scene until at least $M$ actors are labeled. We continue selecting new scenes until the budget is reached.

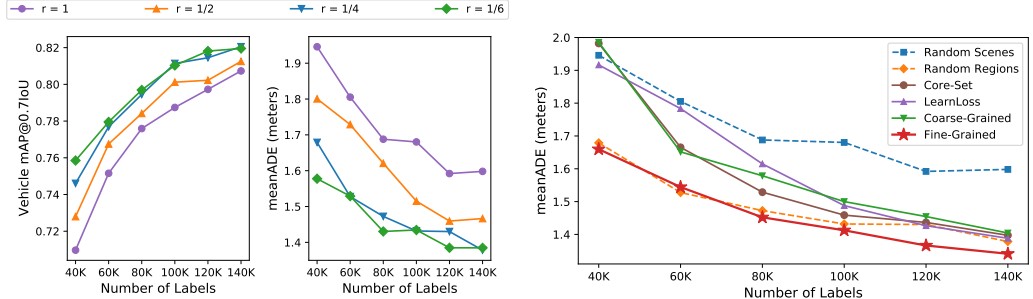

Figure 2: (**Left**) P&P performance when trained on partial labels at varying densities, $r$. (**Right**) Performance of various selection approaches over $N = 5$ active learning iterations.

### 3.3 Selection Criteria

Our fine-grained, cost-aware active learning formulation introduced above generally supports any approach which provides an informativeness score $S(\mathbf{x}_R)$ per example. In this work, we focus on uncertainty-based approaches, an extremely common active learning paradigm based on the assumption that training on uncertain examples are most likely to improve future performance. As this approach depends on a model's characterization of uncertainty, we first describe our probabilistic P&P model, followed by possible uncertainty-measures that can be used for scoring.

**Model:** Following [6], we jointly train a model for both perception and prediction from LiDAR and HD map inputs. A model which naturally characterizes uncertainty over predictions is desirable, as these uncertainty estimates provide a useful measure of informativeness for scoring. Therefore, we leverage the output representation of [11], using a mixture of $K$ Gaussians to represent the distribution of each actor's future positions. For simplicity, independence is assumed between timesteps of the prediction horizon. Thus the likelihood of a particular actor trajectory, $\mathbf{y}_i$, can be written as,

$$p(\mathbf{y}_i) = \sum_{k=1}^{K} \pi_k \prod_{t=1}^{T} \mathcal{N} \left( \mathbf{y}_i; \mu_k^t, \mathbf{\Sigma}_k^t \right) \quad , \tag{3}$$

where $\mathcal{N}$ is the pdf of a 2D multivariate Gaussian with parameters $\mu_k^t$, $\mathbf{\Sigma}_k^t$, and $\pi_k$ represent Gaussian mixture weights. These parameters, for each detected actor, are predicted by a deep neural network trained with negative log likelihood (see Section 4 for more details). With knowledge of the model, we now introduce measures of uncertainty to use as selection criteria. Due to the multi-task nature of the task, we present separate selection criteria for the detection and prediction task. In practice, a mix of both can be used to ensure performance improves across both tasks.

**Detection Entropy:** We focus on characterizing the uncertainty over the model's classification predictions for each anchor. For classification tasks, the uncertainty is typically estimated by calculating the entropy of the model's predicted probabilities. Given anchors $a \in \mathcal{A}$ with associated probabilities $p_a$, the entropy of the predictions are given by,

$$H_D(\mathcal{A}) = - \sum_{a \in \mathcal{A}} p_a \log p_a + (1 - p_a) \log(1 - p_a) \quad . \tag{4}$$

To score regions, we assume independence between anchors and sum the entropies of anchors in $R$.

**Prediction Entropy:** Computing prediction entropy naturally depends on the model's output representation. Our model outputs a Gaussian mixture for each predicted actor. Unfortunately, there is no known closed form solution to computing this distribution's entropy [39]. Therefore, we are required to estimate the entropy via approximations. We explored various approximations, including a sample-based monte-carlo estimate, but all performed similarly to or worse than an approximation via the entropy of the discrete categorical distribution induced by the mixture weights $\pi_k$,

$$H_P(\mathbf{y}_i) = - \sum_{\pi_k} \pi_k \log \pi_k \quad . \tag{5}$$

Intuitively, this approximation is well-suited to capture cases where the model is uncertain between multiple possible modes, which is likely representative of the true entropy of distribution. We use

| Selection | Prediction (meanADE) ↓ | | | | Downstream Planning | | | | |
|---|---|---|---|---|---|---|---|---|---|
| | Straight (m) | Left (m) | Right (m) | Stationary (m) | Collision ↓ (%) | L2 ↓ (m) | Lat. acc. ↓ (m / s$^2$) | Jerk ↓ (m / s$^3$) | Progress ↑ (m) |
| Random Scenes | 2.89 | 5.31 | 5.68 | 0.22 | 5.02 | 5.89 | 2.80 | 2.67 | 33.5 |
| Random Regions | 2.46 | 4.82 | 4.96 | **0.20** | 5.07 | 5.71 | 2.70 | 2.47 | 33.6 |
| Core-Set | 2.45 | 4.71 | 5.01 | 0.21 | 5.14 | 5.72 | 2.65 | 2.45 | 33.6 |
| LearnLoss | 2.46 | 4.74 | 4.99 | 0.21 | 5.15 | 5.74 | 2.68 | 2.47 | 33.6 |
| Coarse-Grained | 2.44 | 4.79 | 5.03 | 0.22 | 5.17 | 5.71 | 2.67 | 2.44 | **33.8** |
| Fine-Grained | **2.29** | **4.52** | **4.91** | 0.21 | **4.63** | **5.56** | **2.62** | **2.38** | 33.7 |

Table 1: (**Left**) Prediction Performance By High Level Action (**Right**) Planning Performance.

this approximation due to its simplicity and computational efficiency while providing similar performance. Finally, we assume independence and sum the actor entropies within each region.

## 4    Experiments

In this section, we analyze the effects of partial labeling and fine-grained active selection. First, we explore partial labeling independent of active selection. Next, we explore the improvements provided by active selection for prediction. We find that a simple prediction entropy combined with fine-grained active selection outperforms various traditional scene-based approaches. More detailed analysis shows that fine-grained selection enables the learner to better oversample labels exhibiting complex driving behaviors, resulting in better performance on these challenging behaviors in the test-set. In practice, we are most interested in the effects of these improvements on the downstream motion planning task, where we find significant improvements across most metrics. Finally, we observe similar improvements for perception when using detection entropy as the selection criterion.

**Dataset:**   We leverage a real-world large-scale dataset collected across multiple cities in North America. To simulate the active learning setting, we follow standard practice in active learning research and treat the large labeled dataset as an unlabeled pool. Active learning approaches select from this pool containing 100K scenes, with roughly 2 million labels. For each scene, we have access to LiDAR sweeps recorded at 10Hz with a localized HD map given as input to the model. For evaluation, we use the standard metrics of mAP@0.7 for detection and meanADE for prediction.

**Implementation Details:**   For model implementation, we follow the exact details of the Gaussian mixture baseline from [12], an implementation of MTP [11] for the joint perception and prediction setting. To support partial labeling, we find it is necessary to use `sum` instead of `mean` to reduce losses in a batch, ensuring actors in less densely labeled scenes are not up-weighted relative to those in more densely labeled scenes. All models are trained for 50 epochs, using budgeted training [40] for the learning rate schedule. We use $H = W = 20$ to discretize the entire scene into 400 rectangular regions for scoring and selection. Finally, for sparsity regularization, we set $M = 5$.

### 4.1    P&P from Partial Supervision

To test the effects of partial labeling, we randomly select labels at varying levels of ground-truth density per scene. To ensure a fair comparison, the total labeling budget is fixed across densities. Specifically, let $r$ be the labeling density. When $r = 1$, we recover the traditional fully labeled setting. When $r = \frac{1}{2}$, we first sample scenes randomly and then sample regions from half of each scene, providing labels only for selected regions. Notice that since we fix the labeling budget, selecting at lower densities $r$, will result in more scenes. Figure 2 shows the effect of partial labeling on P&P performance. Across all dataset sizes, we observe improved performance with lower density datasets. Benefits appear to saturate at $r = \frac{1}{4}$, as the sparser labeling density $r = \frac{1}{6}$ does not provide further improvements. Performance gains can be explained by the fact that more sparsely labeled datasets naturally include supervision from more scenes, improving the model's ability to generalize. Our results suggest adopting sparse labeling to optimize P&P performance under a fixed labeling budget rather than label scenes entirely. However, we note that in practice there is a tradeoff between labeling at lower densities and model training times, as datasets with less dense supervision must be trained for more iterations. We explore this further in our sparsity regularization experiments.

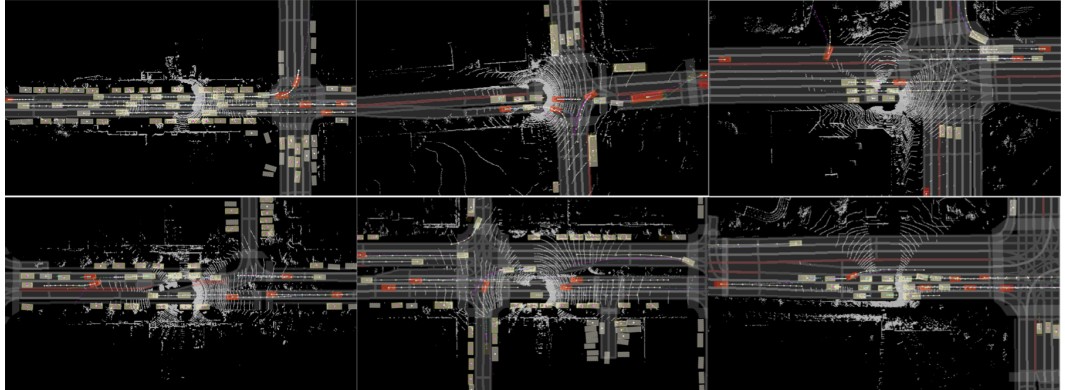

Figure 3: **Qualitative Examples:** Labels in regions selected by fine-grained selection shown in **red**. Selected regions tend to have moving vehicles performing interesting actions (e.g. U-Turns).

## 4.2 Fine-Grained Active Selection for Prediction

**Experimental Setup:** In this experiment, we study active learning approaches for improved prediction. For each method, we sample an initial labeled set $X_L^{(0)}$ of 40K vehicles from $X_U$. To ensure that final datasets contain scenes with similar density of supervision, the initial data for fine-grained methods are partially labeled scenes, whereas coarse-grained approaches sample full scenes. For fair comparison, we fix the labeling budget at each active learning iteration. Specifically, for each of the $N = 5$ active learning iterations, the learner is given a budget of $B = 20$K vehicles. After each iteration, the model is re-trained on its current set of labels and evaluated on a seperate held-out test set. The test-set is held constant across all approaches and contains traditional fully labeled scenes.

**Additional Baselines:** We compare fine-grained active selection to full scenes selected randomly (`Random Scenes`), partially labeled scenes at density $r = \frac{1}{4}$ selected randomly (`Random Regions`), full scenes selected by prediction entropy (`Coarse-Grained`), and two additional active learning baselines adapted to the P&P setting. The first is a recent uncertainty-based approach which learns to predict the loss of unlabeled examples, which we refer to as `LearnLoss` [1]. The second is the common diversity-based approach of `Core-Set` selection [21]. See the supplementary for details.

**Prediction Performance:** Results are shown in Figure 2. All active selection techniques offer significant improvements over random selection. Interestingly, despite large differences in the selection criteria (e.g., uncertainty-based vs. diversity-based), scene-based approaches achieve similar performance, indicating that gains may be saturated due to the inflexibility of selecting entire scenes. Surprisingly, simply labeling random regions appears to perform better than or similar to many of the coarse-grained active learning approaches. Finally, fine-grained selection offers the best performance. While the improvements may appear relatively small, we find consistent results across random seeds due to the large dataset sizes. Additionally, aggregate prediction metrics are averaged over more than 1M actors in the test-set and may hide large differences between the specific behaviors of the prediction models, calling for more detailed analysis.

**Performance By High Level Action:** In Table 1, we break down the prediction performance by action: driving straight, turning left, turning right, and stationary. We notice that differences between selection algorithms become more apparent across actions associated with more difficult predictions (i.e., all non-stationary actions). These results are explained by the fact that fine-grained entropy selects more unpredictable moving actors compared to other selection approaches.

**Planning Performance:** Following [12], we evaluate downstream performance on motion planning, computing the collision rate, L2 error, lateral acceleration, jerk, and progress of a planner [41] given the predictions from each trained model. The results in Table 1 show that fine-grained selection leads to a significant reduction in collisions and also outperforms all baselines across the remaining planning metrics, except progress, for which the differences are not significant (an obser-

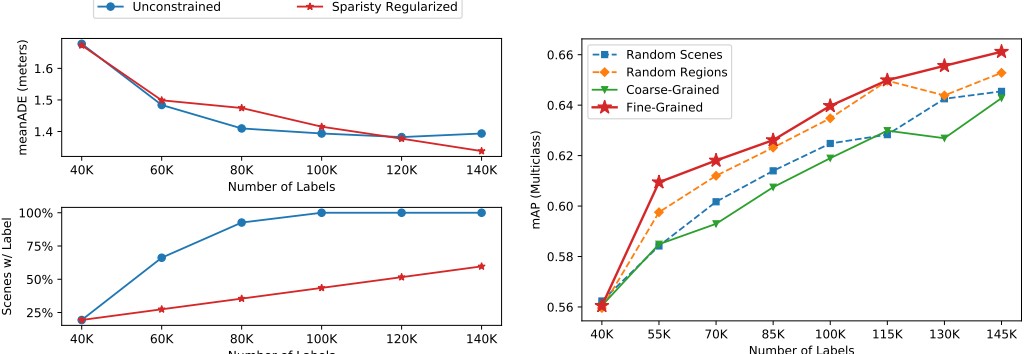

Figure 4: (**Left**) The effect of sparsity regularization on performance (top) and training time (bottom). (**Right**) Multi-class detection performance for $N = 7$ iterations of Active Learning.

vation consistent with [12]). The results demonstrate that the challenging training examples selected by active selection leads to improved planning, which is ultimately most important for self-driving.

**Qualitative Examples and Selection Statistics:** Examples of regions selected by fine-grained selection, seen in Figure 3, tend to include vehicle labels with moving actors, actors at intersections, or actors performing odd maneuvers (e.g., U-Turn in the top-right example). Non-moving actors are rarely selected. In the supplementary materials, we plot histograms of label statistics selected by each method after the final iteration of active learning. As expected, we notice that active-selection methods tend to sample more non-stationary vehicles and vehicles further from the SDV. This effect is more apparent for fine-grained selection methods due to the additional flexibility provided by the partially labeled setup. Please see the supplementary for more detailed analysis.

**Sparsity Regularization Ablation:** In Figure 4, we ablate the effect of sparsity regularization on performance and number of scenes selected. As expected, at early iterations, we find unconstrained selection outperforms the sparsity regularized approach, as the unconstrained approach has the freedom to select a larger set of scenes, each with less supervision. However, at later iterations, we observe the unconstrained selection performance degrades. This is likely caused by the imbalance between the amount of supervision available for each scene, which we found empirically can lead to degraded performance. Beyond performance, there is an additional, perhaps more important, benefit of sparsity regularization. Since the active learner must select at least $M$ actors per scene, the number of scenes in the dataset grows linearly with each iteration. Alternatively, in the unconstrained approach, the dataset size explodes at early iterations until there is a label for every scene in $X_U$.

### 4.3 Fine-Grained Active Selection for Perception

We additionally experiment with fine-grained selection for improved perception. We follow a similar experimental setup, replacing prediction entropy with detection entropy. As all methods perform similarly on vehicle detection, we evaluate on the more challenging multi-class setting where cyclists and pedestrians must be detected. Similar to the prediction setting, results in Figure 4 show fine-grained selection is most effective. Interestingly, coarse-grained selection is similar to random scenes, likely explained by an averaging effect from summing the entropies over the full scene.

## 5 Conclusion

We studied active learning techniques to intelligently select examples to label from unlabeled self-driving data logs for perception and prediction models. We found the traditional active learning setting ill-suited and introduced generalizations to account for variable labeling costs and enable fine-grained selection through partially labeled scenes. In our experiments, we found significant improvements from partial labeling without any active selection, and further gains across perception, prediction and downstream planning by leveraging fine-grained active selection. Our results demonstrate that the dominant paradigm of labeling entire self-driving scenes may not be most efficient under a fixed budget and that fine-grained selection is likely required for maximal efficiency.

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
