# OpenReview forum: "Just Label What You Need: Fine-Grained Active Selection for P&P through Partially Labeled Scenes"
_robot-learning.org/CoRL/2021/Conference — CoRL2021 Poster_

### Official Review · Reviewer_TXdJ · 2021-07-23

**Originality:** Fair
**Technical Quality:** Good
**Clarity Of Presentation:** Very Good
**Impact:** 3

**Recommendation:**

Weak Reject: I recommend rejecting the paper, but will not argue for my recommendation if the majority of other reviewers have a different opinion.

**Summary:**

The authors present a method for fine-grained active learning in the self-driving domain.  At each round of training, a learning agent scores unlabelled examples that seem most informative, and an oracle provides labels for them.  The agent then trains on all labeled examples seen so far and repeats.  The authors argue that performing this active selection on a per-instance basis rather than a per-scene basis improves performance.

**Issues:**

It would be great if you can provide more clarity on what constitutes significant improvement in this domain, and make a strong argument that the improvements here will have a strong impact on real-world performance.

Label/explain the units in Table 1 please.

**Reviewer Expertise:**

Good: General knowledge of the area

**Strengths And Weaknesses:**

Strengths:
The paper is very well written and easy to follow.  Nothing is confusing, and the material is well presented.

The experiments are thorough and the authors have presented good baselines to disentangle the effects of active learning vs. fine-grained/instance-based labelling.

The results seem to provide useful insight for practitioners in this space.

Weaknesses:
There is not a great deal of technical novelty here.  The key idea is to do active learning over instances rather than full scenes, which leads to a fairly straightforward implementation.

The authors heavily promote their approach based on active learning, but the most important result of the paper seems to be that if you have a limited labeling budget, it’s much better to partially label many scenes than densely label a few, regardless of whether you use active labeling or just random selection.  The difference between active fine-grained and random fine-grained (random regions) is quite small in the immediate perception/planning results (Figure 2.)  There is more of a gap for the downstream planning results (Table 1.) but here the gaps between all methods seem relatively small.  In general it would be very helpful if the authors provided some guidance as to what constitutes a significant improvement in this domain.  The authors comment that the difference of 0.1 “Progress” where their model is weaker than the random baseline is not significant, but do not provide information on what threshold would be considered significant here.  I understand that with self-driving cars every collision counts, but I also have a limited understanding of how well performance on these held-out test sets relates to real-world performance.  More explanation here would be helpful.

Along these lines, please provide units for the various error rates in Table 1 to give a better idea of what these numbers mean.


**Summary Of Recommendation:**

I am very much on the fence with regards to this paper.  It is well presented, and well argued, but the minimal technical novelty and results which are marginally better, but not dominating other methods makes it hard to strongly argue in favor of acceptance.  Part of this stems from my own lack of familiarity with what constitutes a significant improvement in this domain.  I would improve my score if the authors can convincingly argue that these results are significant and useful and/or other reviewers can vouch for their significance.

Also, if there was an option for Impact between 3 and 4 (something like "This work is likely to have some impact within a limited domain") I would apply it here.  It is reasonable to see this paper having some impact within the self-driving community.

---

> ### Author Response · Authors · 2021-08-26
> **Response to Reviewer TXdJ**
>
> Thank you for your questions and helpful feedback! Please see our responses below.
>
> **Q1) On lack of technical novelty**
>
> A1) We respectfully disagree that our approach lacks novelty. Our work explores active learning in a novel problem setting and introduces a cost-aware, fine-grained selection strategy to overcome the shortcomings of prior approaches in this domain. Our approach requires introducing a new method for training P&P models from partial supervision, which was non-trivial to implement successfully. For example, we encountered challenges when training with datasets with imbalances in the area of labeled regions across examples, which led us to propose our sparsity regularization technique to ensure better compute tradeoffs and improved performance at later AL iterations (see Figure 4 of our paper). Finally, the novelty and contributions of this work are recognized by other reviewers and the AC. Specifically, Reviewer Pw9h believes it is a “novel idea to explicitly model the cost and incorporate it in the active learning selection process” and the AC believes that our work can “generalize to other domains as well.”
>
> **Q2) On significance of the planning results**
>
> A2) We appreciate the reviewer’s concerns that it is difficult to assess whether the planning improvements are significant. When analyzing these metrics, it is important to keep in mind the long-tail distribution of driving behavior. Collisions are rare, but have dire consequences, so significant improvements may appear to be small. From our experience and prior works, the reduction in collision rate from random regions to active selection of 0.44 is significant, especially when considering that the improvements come only from changes to the prediction model from active selection. In prior published work [1], **a similar reduction of 0.46 in the collision rate is considered to be significant (see Table 3 of [1])**. Our improvements in the additional metrics of L2, lateral acceleration, and jerk are also similar to those reported in [1]. Previous works [1, 2] have also observed similar noise when evaluating the planner based on “progress”. All together, the planning metrics indicate our approach leads to a **significant** reduction in collisions, more human-like driving (as measured by L2), improved comfort (i.e. lower jerk/lateral acceleration), with negligible changes to the SDV’s ability to progress along the route. We will make this analysis of the metrics clearer in the revised paper.
>
> **Q3) On units for the error rates in Table 1**
>
> A3) We apologize for the missing units in Table 1. We are revising the paper to include this.
>
> **References**
>
> [1] Casas et. al “Implicit Latent Variable Model for Scene-Consistent Motion Forecasting”, 2020.
>
> [2] Sadat et al. “Perceive, Predict, and Plan: Safe Motion Planning Through Interpretable Semantic Representations”, 2020.

---

> > ### Comment · Reviewer_TXdJ · 2021-09-01
> > **Update**
> >
> > Given the generally positive notes by the other reviewers, and the reference for what is/has been considered significant improvement in this domain in the past (thanks for that), I have no problem bumping this up to a weak accept.

---

### Official Review · Reviewer_KVfT · 2021-07-23

**Originality:** Good
**Technical Quality:** Fair
**Clarity Of Presentation:** Very Good
**Impact:** 3

**Recommendation:**

Weak Accept: I recommend accepting the paper, but will not argue for my recommendation if the majority of other reviewers have a different opinion.

**Summary:**

Self-driving vehicles require extensive data supervision in order to generalize in the real world. However, much of this labelling is wasted effort due to the "long-tail problem" - common situations (e.g. parked car) are easy to predict and make up the bulk of the dataset while rare (but crucial) situations make up a tiny fraction of the data-points. This paper proposes an active learning technique to make maximal use of fixed labelling budgets. The utility $S(r)$ and the cost $C(r)$ of each un-labelled region $r$ are explicitly modeled, and data points are chosen for labelling if they are expected to improve prediction performance most efficiently. Furthermore, the training loss is modified to model partially labelled scenes. These modifications enable improved performance v.s. baselines and strongly indicate that fully labelling scenes is a wasteful process.

**Issues:**

* Please show comparisons v.s. baselines you mentioned as well as any other active learning baselines that have been previously tried in this space.
* Please do a better job documenting the dataset. Did you collect it or did someone else? If you collected it, what sort of data distribution should we expect?

**Reviewer Expertise:**

Fair: Some knowledge of the area

**Strengths And Weaknesses:**

The paper is well written and the big ideas behind the work are effectively conveyed. The supplementary video is very helpful for understanding the paper and the qualitative examples help contextualize the work's contributions. The general idea of using active learning and partial labelling seems like a sensible way to attack the long-tail problem. Furthermore, the ablations do a good job of justifying hyper-parameter choices (e.g. scoring function parameterization and selection policy) in the method.

However, the results section could use comparisons to real active learning baselines. The authors mention *LearnLoss* and *Core-Set* but show no results on those baselines, other than vague references of similarity to *Coarse-Grained* in the supplementary material. Real comparisons to prior active learning/labelling work will help us judge this paper's contributions fairly.

**Summary Of Recommendation:**

This paper is well executed and thorough investigation of partial labelling and active learning in P&P modeling context. However, it is missing comparisons against real baselines that are important to understand significance.

---

> ### Author Response · Authors · 2021-08-26
> **Response to Reviewer KVfT**
>
> Thank you for your questions and helpful feedback! Please see our responses below.
>
> **Q1) On comparisons to Active Learning baselines**
>
> A1) Please see (Q2/A2) in our general response. Additionally, we can appreciate the reviewer’s request for “other active learning baselines that have been previously tried in this space”. Unfortunately, to the best of our knowledge, there is not a single published active learning baseline on the perception and prediction task. Therefore, we chose two strong, general active learning baselines to compare against. There has unfortunately been a lack of study within this domain, likely due to the complexity of the task, and the dataset and model sizes required, which make active learning experimentation challenging. We hope our work inspires future research within this domain.
>
> **Q2) On documenting the dataset**
>
> A2) We apologize for a lack of details on the dataset. The dataset is a large-scale, real-world self-driving dataset collected specifically for the perception and prediction tasks. The dataset is made up of challenging urban driving scenes. We have not disclosed the full details for concerns that it may compromise anonymity, but we will ensure that our final revision contains all relevant dataset details.

---

### Official Review · Reviewer_PW9h · 2021-07-23

**Originality:** Good
**Technical Quality:** Very Good
**Clarity Of Presentation:** Very Good
**Impact:** 4

**Recommendation:**

Weak Accept: I recommend accepting the paper, but will not argue for my recommendation if the majority of other reviewers have a different opinion.

**Summary:**

This paper presents a way to introduce active learning in the Perception and Prediction (P&P) setting: it chooses to label the scenes partially instead of label the entire scene for a higher cost. It trains with partial supervision and only apply the loss on the labeled regions; essentially it will select a large number of scenes, each labelled very sparsely. The methods has been shown effective in P&P tasks and outperform prior baselines.

**Issues:**

Please refer to the Strengths and Weaknesses section for more details.

**Reviewer Expertise:**

Fair: Some knowledge of the area

**Strengths And Weaknesses:**

Strengths:
- It is a novel idea to explicitly model the cost and incorporate it in the active learning selection process.
- Comprehensive literature review
- Clear presentation of methods and algorithms


One concern I have on visualization is that in Figure 3: Qualitative Examples, the graphics do not seem to be very clear in presentation. The red boxes are quite small and I'm not sure are the white boxes meaningful? There could be more semantically meaningful explanations for the figures (although I like the idea of using qualitative examples).

Another question I have is in Figure 2 (Right): the experimental results of Fine-Grained is not too different from that using Random Regions, especially when using small amount of labels. Do the authors have an explanation on the similarity of results?

**Summary Of Recommendation:**

I would recommend this paper since it presents an interesting active learning method in P&P setting that involves labelling cost, and introduces the idea of modularity in training samples. The main idea could potentially inspire active learning in other fields, for example image classification, robot learning etc.

---

> ### Author Response · Authors · 2021-08-26
> **Response to Reviewer PW9h**
>
> Thank you for your questions and helpful feedback! Please see our responses below.
>
> **Q1) On Figure 3 visualization**
>
> A1) Thank you for the feedback on our visualization. The white boxes represent the labeled examples in regions that were not selected by the active learning algorithm. We believe showing these boxes for the entire scene provide important context for the reader as to which regions are selected by the algorithm. Our supplementary video contains more detailed explanations of qualitative examples, in which we emphasize interesting observations within the qualitative examples. Please let us know if you have additional suggestions as to how we can improve the presentation of these examples.
>
> **Q2) On the similarities of fine-grained selection vs. random-regions**
>
> A2) As we argue in the paper, aggregate prediction metrics are not a good measure of the final performance of a self-driving system. We demonstrate in the paper that fine-grained active selection oversamples more challenging vehicle behaviors (please refer to the selection statistics in the supplementary materials) compared to random selection. In Table 1, we show that as a consequence, fine-grained selection leads to more significant improvements on predicting challenging vehicle behaviors whereas "random regions" leads to marginally better performance on stationary vehicles. Since the test set is dominated by stationary vehicles, this leads to similar performance on the aggregate prediction metrics in Figure 2 (right). However, as we show in the paper, the improvements on more challenging behaviors from fine-grained active selection leads to significant improvements in planning performance (see Table 1 of our paper), which is most important for the real-world performance of the self-driving system.

---

> > ### Comment · Reviewer_PW9h · 2021-09-03
> > **Thank you for your response**
> >
> > Thank you for your detailed response! I hold my original opinion of acceptance for the final review.

---

### Official Review · Reviewer_TDyJ · 2021-07-26

**Originality:** Fair
**Technical Quality:** Good
**Clarity Of Presentation:** Very Good
**Impact:** 3

**Recommendation:**

Weak Accept: I recommend accepting the paper, but will not argue for my recommendation if the majority of other reviewers have a different opinion.

**Summary:**

The paper proposes a labeling method for active learning for self-driving applications. In particular, the paper focuses on the percpetion and prediction problem. Unlike traditional active learning frameworks that assume each example in the dataset can be labeled for the same cost, the proposed method explicitly model the cost to label each example. The motivation is, given a fixed labeling budget, more examples (lower cost examples) can be labeled. The authors proposed to include the cost to label in the scoring function, the function used in active learning frameworks to pick examles that are believed to be most likely to improve the model performance.
Furthermore, the paper introduces partial labeling, i.e., labeling only particular regions of an example. The authors show that given a fixed labeling budget, the performance will improve by training with more examples but equal number of labels.

**Issues:**

* An important contribution of the paper is the "cost-aware" part of it. However, this component is not described well and it is not really clear how the cost of each example is predicted.

* For all experimental results, a confidence interval measure is required to ensure statistical significance.

* Line 117: NMS is not defined.
* Line 166: The sentence is incomplete

**Reviewer Expertise:**

Good: General knowledge of the area

**Strengths And Weaknesses:**

Strengths
* The paper is well-structured and the results support the claims.
* The idea is quite simple but seems to have a considerable effect on the results

Weaknesses
* The numbers/results provided lack measures like confidence intervals to ensure statistical significance.
* One of the core contributions of the paper, i.e., predicting the cost of labeling for each example, is not clearly described.

**Summary Of Recommendation:**

It is not really clear how partial labeling would work for the prediction problem. An actor can leave the region in subsequent frames, therefore, the labeling region can change across consecutive frames.

---

> ### Author Response · Authors · 2021-08-26
> **Response to Reviewer TDyJ**
>
> Thank you for your questions and helpful feedback! Please see our responses below.
>
> **Q1) On confidence intervals**
>
> A1) Please see (Q1/A1) in our general response.
>
> **Q2) On cost prediction**
>
> A2) We apologize for the confusion. In line 111 of our paper, we argue that the “labeling cost for P&P examples can be accurately modeled as a linear function of the number of actors in the scene as most annotation time is spent drawing detailed bounding boxes for each actor”. This is based on both real-world experience and the pricing schemes of many 3rd party labeling platforms, which charge for each cuboid annotation, rather than each scene. In line 116, we explain that since this cost is unknown prior to labeling, we estimate it based on the number of detections within a region output by our current model. To summarize succinctly, we estimate the labeling cost of a region as the cost per annotation multiplied by the number of model detections present within that region.
>
> **Q3) On partial labeling for prediction supervision**
>
> A3) This is an excellent point! Modern labeling tools for self-driving scenes employ vehicle motion models which predict the future positions of the vehicle, allowing the annotator to simply make small adjustments rather than re-label the full bounding box in each frame. This same approach could be used to ensure that we adjust the labeling region in future frames to account for the motion of the object. Therefore, we assume that if a vehicle is present within the labeling region at the detection time, we also receive its labels for the prediction horizon, even if the vehicle leaves the labeling region.
>
> **Q4) On NMS**
>
> A4) We apologize for using the acronym NMS, without properly defining it as “Non-Maximum Suppression”. This is a common post-processing algorithm used in object detection to remove duplicate detection outputs by greedily removing repeated detections with lower-confidences [1].  We have revised the paper to include the definition and a reference.
>
> **Q5) On the incomplete sentence**
>
> A5) Thank you for pointing out this sentence, we have revised the paper to correct it.
>
> **References**
>
> [1] Felzenszwalb et al. "Object detection with discriminatively trained part-based models."

---

> > ### Comment · Reviewer_TDyJ · 2021-09-06
> > **Final vote**
> >
> > Thanks for your reply. I will also keep my original score "weak accept".

---

### Author Response · Authors · 2021-08-26
**General Response to Common Concerns**

We thank all reviewers and the AC for their thoughtful comments. We are pleased to see that there is consensus that our paper is both interesting and well-presented.  We were especially happy to see reviewers think our proposed approach is “novel” [Reviewer Pw9h], has a “considerable effect on the results” [Reviewer TDyJ], and could “generalize to other domains as well” [AC]. In what follows, we address common concerns raised by the reviewers.

**Q1) On statistical significance [AC, Reviewer TDyJ, Reviewer TXdJ]**

A1) From our experience within this domain, we strongly believe the results are statistically significant. Unfortunately, due to the computational requirements, it is very expensive to obtain confidence interval estimates of each point on every active learning curve. Please bear in mind that each point on the curve represents a complete training run from scratch on a large-scale self-driving dataset (e.g., a single curve can take over 350 GPU hours to run). Therefore, to address the reviewers’ concerns with a reasonable computational budget, we compare an additional run only for the two approaches with the most similar prediction performance in Figure 2 (right): random regions vs. fine-grained active selection. Starting from a randomly initialized labeled pool different from that used in the main paper, we run 6 iterations of active learning for both approaches, selecting 30K new labels at each iteration. The results, shown below and added to our supplementary materials, are consistent with our original experiments in Figure 2 (right), in which fine-grained selection outperforms random regions across all active learning iterations, demonstrating that it is extremely unlikely that noise is responsible for the difference in performance.

Number of Labels |  Random Regions (meanADE) | Fine-Grained (meanADE) |
--- | --- | --- |
40K     | 1.64   |   1.64  |
70K     |  1.48  | 1.44 |
100K   | 1.40 | 1.39 |
130K   | 1.35 | 1.33 |
160K   | 1.33 | 1.29 |
190K   | 1.31 | 1.25 |

**Q2) On comparisons to other Active Learning baselines [AC, Reviewer KVfT]**

A2) Reviewer KVfT commented that “The authors mention LearnLoss and Core-Set but show no results on those baselines, other than vague references of similarity to Coarse-Grained in the supplementary material. ”. However, **we included the comparisons with these two baselines (LearnLoss and Core-Set) on the prediction and planning tasks in Figure 2 (right) and Table 1**. As we can see, our fine-grained selection approach outperforms prior active learning works approaches consistently in prediction and planning tasks.

---

### Meta-Review · Area_Chair_q8Az · 2021-08-13

**Recommendation:** Accept (Poster)
**Confidence:** 4

**Metareview:**

All reviewers agree that the idea proposed in the paper is interesting and that the paper is well presented. Three of four reviewers rate the paper's impact to be limited, although I can see this to generalize to other domains as well. There are two issues with the paper in its current form though: I agree with reviewer TDyJ who criticizes the lack of confidence intervals to understand the statistical significance of the results. Second, reviewer KVfT recommends a comparison with prior active learning work - something I would love to see as well. If these issues were to be addressed I would recommend acceptance.

---

> ### Author Response · Authors · 2021-08-26
> **Response to Area Chair q8Az**
>
> We thank the meta-reviewer for the helpful summary. Please see (Q1/A1) and (Q2/A2) in our general response which address the two issues raised.

---

### Decision · Program_Chairs · 2021-09-13

**Decision:**

Accept (Poster)

**Comment:**

All reviewers agree that the idea proposed in the paper is interesting and that the paper is well presented. Three of four reviewers rate the paper's impact to be limited, although I can see this to generalize to other domains as well. There are two issues with the paper in its current form though: I agree with reviewer TDyJ who criticizes the lack of confidence intervals to understand the statistical significance of the results. Second, reviewer KVfT recommends a comparison with prior active learning work - something I would love to see as well. If these issues were to be addressed I would recommend acceptance.